# Implementing Nitrogen Vacancy Center Quantum Sensor Technology for Magnetic Flux Leakage Testing

**DOI:** 10.3390/s25237279

**Published:** 2025-11-29

**Authors:** Jonathan Villing, Matthias Niethammer, Luca-Ion Arişanu, Frank Lehmann, Harald Garrecht

**Affiliations:** 1Materials Testing Institute (MPA), University of Stuttgart, 70569 Stuttgart, Germany; frank.lehmann@mpa.uni-stuttgart.de (F.L.);; 2Advanced Quantum GmbH, 70178 Stuttgart, Germany; m.niethammer@advanced-quantum.de (M.N.); l.arisanu@advanced-quantum.de (L.-I.A.)

**Keywords:** quantum sensors, magnetic flux leakage, nitrogen vacancy centers, prestressing steel, non-destructive testing, structural health monitoring

## Abstract

Ensuring the structural integrity of prestressed (PS) concrete is essential for the safety and longevity of infrastructure. Magnetic Flux Leakage (MFL) testing is a widely used non-destructive testing (NDT) method for detecting fractures in prestressing steel. This study explores the application of quantum sensors based on nitrogen vacancy (NV) centers in artificial diamonds for MFL testing and presents a novel method for processing continuous-wave optically detected magnetic resonance (CW-ODMR) data into vectorized magnetic field measurements. These sensors offer high sensitivity, low hysteresis, and multi-directional magnetic field detection, making them a promising alternative for advanced NDT applications. A data processing framework was developed to transform CW-ODMR measurements into vectorized magnetic flux density values in the x, y, and z directions. This process enables the conversion of crystallographic sensor orientations into calibrated field directions, ensuring precise magnetic field reconstruction. The method was validated through 121 fracture measurements and 19 open-bar-end measurements, demonstrating its effectiveness in extracting high-resolution vectorized magnetic field data. A subsequent statistical evaluation quantified the influence of sensor displacement, magnetization direction, magnetization distance, and measurement distance. These findings establish a foundation for integrating quantum sensors into MFL-based NDT, with potential applications extending beyond building inspections to a wide range of advanced sensing technologies in scientific and industrial fields.

## 1. Introduction

Sports halls, industrial buildings, railway stations, airports and bridges all have one thing in common—they need to be maintained. Globally, infrastructure and engineering structures are increasingly susceptible to deterioration [1]. Effective maintenance necessitates targeted repairs to address specific deficiencies. To identify these flaws, a wide range of non-destructive testing (NDT) methods has been developed. Each method offers distinct advantages, limitations, and application constraints, highlighting the importance of selecting the most suitable approach for each scenario [2,3,4].

A significant proportion of infrastructure and engineering structures are built using prestressed (PS) concrete, a material known for its ability to span large distances with durability, structural strength, and aesthetic appeal [5,6]. However, a critical failure mode is the rupture of PS tendons, which provide substantial load-bearing capacity and can cause sudden structural collapse [7]. Although tendons are typically safeguarded by surrounding concrete and mortar, design flaws or construction defects can compromise stability. In older structures, the absence of modern design standards further elevates this risk [8].

Common deficiencies include insufficient concrete cover, inadequate grouting, cracking, mechanical damage, fatigue, overloading, and the use of alumina cement or quenched and tempered PS steel susceptible to hydrogen-induced corrosion [8,9]. Notably, in certain PS structural designs, damage may not manifest as surface cracks before failure, rendering it undetectable through conventional building inspections [9]. In most cases of abrupt collapse, hydrogen-induced stress corrosion cracking was identified as the primary failure mechanism [10].

Various NDT methods have been explored to detect fractures in PS steel, with magnetic techniques showing the greatest potential for inspecting large areas of PS concrete structures [7,11,12,13]. Among these, Magnetic Flux Leakage (MFL) testing is particularly effective in identifying fractures [9,14]. This method involves magnetizing the steel using an external magnetic field and measuring the resulting MFL field with magnetic field sensors.

MFL testing can be configured in several ways, depending on the choice of magnet and measurement timing. The magnet can be either a permanent magnet or an electromagnet, and the MFL signal can be measured either during magnetization or in remanence [14,15]. Both measurement options are used in industrial applications [16]. Some systems only measure during magnetization, which is mainly the case with permanent magnet configurations [17,18]. Other systems rely primarily on remanent measurements [9] and others combine both approaches [13,15]. A range of magnetic field sensors are available for this purpose, each varying in sensitivity, dynamic range, hysteresis, resistance to environmental conditions, and cost [19]. Selecting the appropriate configuration and sensor type is crucial for accurate and reliable fracture detection [20].

Hall-effect sensors are widely used in MFL applications due to their versatility, lack of hysteresis, low power consumption, linearity, cost-effectiveness, and compact size. However, their sensitivity is limited. State-of-the-art laboratory Hall-probes can reach up to nT/√Hz range at room temperature. The sensitivity of standard Hall-effect sensors does not reach that level of sensitivity in practical applications. Additionally they can only measure magnetic fields in a single direction [21]. Multi-directional measurements are achieved by arranging Hall sensors in arrays with varied orientations [20,22]. However, this approach compromises spatial resolution and sensitivity, presenting limitations in certain applications. Induction coil magnetometers offer higher sensitivity, reaching pT/√Hz to fT/√Hz depending on frequency and coil size. They have low hysteresis, but require large coil sizes, rendering them impractical for building inspections [23]. More sensitive alternatives include fluxgate magnetometers, magnetoresistive (MR), and Giant-magnetoresistive (GMR) sensors, which provide sensitivity in the pT/rtHz range [20]. However, their usage is constrained by hysteresis and nonlinearity challenges [21,24]. Superconducting Quantum Interference Device (SQUID) magnetometers offer superior sensitivity in the aT/rtHz range with minimal hysteresis but necessitate cryogenic cooling to approximately 4 K (low-Tc) or 77 K (high-Tc) depending on design, limiting their field applications. Requiring a cryogenic engine also significantly increases the price [21]. Optically pumped magnetometers (OPMs) in alkali-metal vapors such as cesium or rubidium. These alkali metal vapor optical sensors achieve sensitivities in the range of fT/√Hz at room temperature with millimeter-scale sensing volumes [25]. While most OPMs are scalar devices that measure the magnetic-field modulus, vector-measurement schemes have been demonstrated [26]. An additional option is the use of multi-channel systems [25]. Their advantages include high sensitivity and compactness; however, their dynamic range and gradient tolerance are limited [27,28,29]. The price of a fully functioning sensor setup is higher than conventional sensors, but still more affordable than SQUID magnetometers [25]. Quantum sensors based on negatively charged nitrogen vacancy (NV) centers achieve sensitivities from pT/√Hz to nT/√Hz and also have a high dynamic range up to several mT. In addition, they enable simultaneous measurement of magnetic fields in multiple directions, offering a promising alternative for advanced MFL testing. They operate at room temperature and exhibit low hysteresis. They are also compact and robust, enabling their use on construction sites [24,30].

Various MFL measurement systems have been developed using different configurations of sensors and magnets. Kusenberger et al. utilized a DC current-excited electromagnet and Hall-effect sensors mounted on an inspection cart that moved along a beam using track attachment brackets or movable transverse support members [31]. Fernandes designed a system with an electromagnet and Hall-effect sensors positioned between the poles for MFL measurements and on the pole faces for main magnetic flux (MMF) measurements [5,11,32]. Vogel et al. investigated MFL testing for fatigue break detection in bridges using a triaxial magneto-inductive sensor (MicroMag3) combined with a permanent magnet assembly of five Neodymium Iron Boron (NdFeB) discs [33,34,35].

Fraunhofer IZFP collaborated with Walther to develop the BetoFlux system, which initially focused on detecting corrosion in PS concrete pylons. The first version used a permanent yoke magnet and an array of 32 Hall-effect sensors [17]. A later version, designed to inspect all conventional PS tendons in concrete structures, featured permanent magnets with a flux bridge and an expanded array of 60 Hall-effect sensors [6,36].

Al Ghorbanpoor et al. introduced the magnetic field disturbance system in 1981, utilizing an electromagnet and four Hall-effect sensors. In 2001, he enhanced it to a modular system with an array of 10 Hall-effect sensors for transverse and axial magnetic field measurements [37]. In 2016, he developed a fully robotic system for inspecting PS concrete I-girders, incorporating two NdFeB permanent magnets and a sensor array of 64 one-dimensional Hall-effect sensors arranged in 32 pairs for axial and transverse magnetic field components [18].

Hillemeier et al. created various MFL systems, including REM40 (30 kg), REM150 (200 kg), and REM350 (3000 kg), tailored for specific measurement objectives. These systems employed different sizes of electromagnets and Hall-effect sensor arrays [38]. An innovative feature in REM200 and REM350 is the rotational scanner, which uses ten Hall-effect sensors mounted on a rotating mechanism, enabling dynamic synchronization and measurement of the transverse magnetic flux density [39].

SQUID sensors have been the only effective alternative for MFL measurements in fracture detection. Krause and Sawade developed a system combining a yoke electromagnet, four SQUID magnetometers, and a cryogenic engine to cool the sensors to 78 K [21]. Although SQUID sensors demonstrated superior sensitivity and performance compared to Hall-effect, Fluxgate, GMR, and MR sensors, their complex measurement procedures and cooling requirements ultimately favored the use of Hall-effect sensors in practical building inspections [12,15,21,40,41].

Quantum sensors based on NV centers exhibit high sensitivity while operating at room temperature. Additionally, they can simultaneously measure multiple directions of the magnetic field within a single measurement, enhancing their effectiveness in complex magnetic field analyses. Provided that the bias field is optimized for the application, they have a high dynamic range and gradient tolerance. They have been utilized in laboratory settings for some time; however, only recent technological advancements have enabled their application in real-world scenarios. As a novel sensor type in NDT measurements, they exhibit promising properties for a broad range of magnetic methods. Nevertheless, their high sensitivity and complex data structures present challenges in data interpretation and practical implementation.

Overcoming these challenges requires innovative approaches to data analysis and system integration. This paper evaluates Quantum sensors based on NV centers, demonstrating their effectiveness in MFL testing for fracture detection in PS steel.

## 2. Magnetic Flux Leakage Testing for Fracture Detection in Prestressing Steel

Magnetic flux leakage (MFL) testing relies on specific physical properties, particularly magnetism. There are three types of magnetism: diamagnetism, paramagnetism, and ferromagnetism. However, the effects of diamagnetism and paramagnetism are negligible compared to ferromagnetism, making MFL testing primarily applicable to ferromagnetic materials [42]. Under room temperature, only iron, nickel, and cobalt exhibit ferromagnetic properties, and these elements constitute a significant portion of steel, a critical building material [43].

In their natural state, ferromagnetic materials do not emit a magnetic field, as their internal magnetic moments are randomly oriented, canceling each other out. When exposed to an external magnetic field, these moments align with the field direction, increasing the material’s magnetic field until saturation is reached [42,43].

The resulting magnetic field forms a distinct 3-dimensional pattern around the ferromagnetic object, influenced by the orientation of magnetic moments and the object’s shape. For simple geometries, such as unbroken PS steel or single fractures, this magnetosphere is predictable according to physical laws [39,44]. Figure 1 shows a sketch of the measurement procedure of MFL testing for fracture detection in PS concrete structures and the resulting magnetic stray fields of a fracture.

During MFL testing, the steel is magnetized using an external magnetic field, generating a magnetic leakage field. Fractures in the prestressing steel create a magnetic dipole, locally altering the magnetic stray field. Magnetic field sensors detect these anomalies, enabling fracture identification [15,32]. However, in real-world scenarios, numerous influencing factors interact, complicating the analytical calculation of the magnetosphere. Physical simulations and experimental data are essential for accurate modeling and visualization [39].

When testing real structures, ferromagnetic elements within PS concrete can be magnetized, generating stray fields. The objective is to detect fracture signals, while all other influences are considered noise. Notably, reinforcement elements often produce stronger signals than PS steel fractures. The main reason for this is the distance between the ferromagnetic element and the sensor. Reinforcing bars in particular are usually the element closest to the sensor and therefore generate the strongest signal. Discriminating fracture signals from noise requires an in-depth understanding of the magnetic properties of all relevant elements [15,44].

Another factor that complicates discrimination of fracture signals from noise is specific arrangements of supplemental reinforcement and embedded steel components that produce magnetic fields that closely resemble fracture signals [10,44,45]. This complex signal-to-noise ratio is an inherent physical phenomenon, independent of the sensor type used.

## 3. Quantum Sensors Based on Nitrogen Vacancy Center

Quantum effects were historically observed as random laboratory phenomena. However, with advanced techniques, they can now be initialized, manipulated, and measured with precision and repeatability. A significant advancement in MFL measurement is the development of quantum sensors for magnetic field detection. Technological progress has enabled the miniaturization and robustness of these sensors, making them suitable for field applications, including construction sites. These quantum sensors utilize negatively charged nitrogen vacancy (NV) centers within the crystal lattice of artificial diamonds, leveraging electron spin defects for precise optical magnetic field analysis.

Artificial defects within the diamond’s crystal lattice, known as NV centers, are created by substituting a nitrogen atom adjacent to a vacancy. These point defects form three-dimensional potential wells, introducing discrete energy states within the electronic band structure. Optical transitions between these states result in unique absorption and fluorescence properties. NV centers can adopt four crystallographic orientations within the diamond lattice. By utilizing an ensemble of NV centers with uniformly distributed orientations, complete vector magnetic field characterization can be achieved [46].

NV centers can be utilized for sensing either as single defects or as an ensemble of defects, known as ensemble sensing. Single NV setups offer superior spatial resolution but have lower sensitivity and higher setup costs. Conversely, ensemble sensing provides high sensitivity at a lower cost, albeit with reduced spatial resolution. In MFL testing for fracture detection, sensitivity is prioritized over spatial resolution, making ensemble NV centers the preferred choice. Figure 2 illustrates the optical system embedded in the sensor head.

The sensor head consists of the following elements:a diamond with an ensemble of NV centersthe optical elements to collimate and focusgreen lasera photodiode and amplifiera microwave (MW) antennatransimpedance amplifier

All the necessary electronics are included in the control unit with the following elements:microwave signal generatormicrowave amplifieranalog to digital convertermeasurement control processor

We use a 525 nm green laser to optically pump the NV centers into excited states. The system can then decay back to the ground state through two pathways. The first one is the fluorescence pathway, where the electrons rapidly relax to the ground state within nanoseconds. The emission shows a zero-phonon line (ZPL) at 637 nm. Additionally, the second pathway involves intersystem crossing (ISC) to a metastable singlet state, leading to an effectively non-radiative decay process. The resulting photons are detected by a photodiode, with optical filters used to separate the excitation light from the fluorescence photons.

Additionally, the absorption and emission spectra are influenced by lattice vibrations, known as phonons. The specific vibrational modes, or phonon modes, broaden the optical absorption and thus allow for non-resonant excitation. In the emission, the spectrum is broadened by the phonon side-band to a spectral range between 630 nm to 750 nm.

When the laser is activated, fluorescence produces a steady stream of photons. By continuous excitation, the state is initialized by optical cycling into the bright state. Optical filters ensure that only fluorescence photons within the 600 to 800 nm range are detected by the photodiode. If the quantum state is manipulated to favor ISC, fewer photons are detected. This manipulation is achieved using an MW signal.

A continuous sweep of MW frequencies is applied at the NV center’s location, typically centered at 2.87 GHz, generating a symmetric spectrum around this frequency. The MW frequency range determines the maximum observable magnetic field magnitudes. Light emission is recorded for each MW frequency, forming the Optically Detected Magnetic Resonance (ODMR) spectrum. This technique, known as continuous-wave ODMR (CW-ODMR), involves the continuous application of both the optical excitation laser and MW signals to the NV centers.

## 4. Experimental Setup

Research on MFL measurement for detecting PS steel failures began in the 1980s, leading to extensive experimentation on various failure modes and interference from surrounding rebars. However, the introduction of ODMR measurements necessitates the repetition of many of these experiments to accurately interpret the new data structures and improve fracture detection. The experimental setup is specifically designed to explore the fundamental mechanisms underlying fracture detection in PS steel.

### 4.1. Magnetization System

A permanent magnet system was chosen for magnetization due to its advantages of low weight, compact size, zero power consumption, and cost-effectiveness. The simulation phase aimed to determine the optimal magnet configuration that generates an appropriate field profile and sufficient magnetic flux density.

To achieve this, various configurations of magnet assemblies were systematically evaluated using numerical simulations. Given the traditional use of U-shaped yoke magnets with end-positioned poles, the simulations focused on different U-shaped permanent magnet arrangements. The target magnetizing field at the PS steel bar (≈10 cm below the concrete surface) is approximately 100 G. The simulations aimed to identify a configuration that provides sufficient field strength while meeting structural constraints and practical requirements.

By varying the geometric arrangement of the magnetic components, the impact of key design parameters on the field distribution was examined. The results indicate that specific configurations enhance the field magnitude, while others lead to diminishing returns beyond a certain threshold. The optimal arrangement was determined by maximizing the generated field while maintaining structural feasibility and stability.

Based on the findings, an assembly consisting of fifteen neodymium permanent magnets was selected. It comprises three identical U-shaped structures arranged side by side, each incorporating magnets strategically oriented to optimize the magnetic field. The simulated spatial distribution of the resulting magnetic field is illustrated in Figure 3, confirming that the chosen assembly produces a field profile comparable to that of a conventional yoke electromagnet.

Simulations indicate that the magnetic field strength is sufficient for effective magnetization within the target measurement range. The field remains uniform around the central measurement position, ensuring consistent magnetization of the steel bar. To maintain structural stability, a supporting framework was designed to counteract the strong magnetic attraction forces between components, securing the magnets in the desired configuration.

### 4.2. Sensing System

The NV sensor, developed by Advanced Quantum GmbH (Stuttgart, Germany) consists of a sensor box and a cylindrical diamond sample enclosure. This arrangement is shown in Figure 4. The diameter of the sensor head is 35 mm, with a length of 12 mm, including the laser driver, detector, optics and TIA. The dimensions of the sensor element are 0.5 mm * 0.5 mm * 0.5 mm. Precise measurement of the sensitivity has not been conducted; however, it is estimated to be within the range of 1 nT/rtHz. Linearity range: In the measurement mode with post-processing that was utilised, the capacity is constrained by the microwave source. In this system, it is approximately 7.5 mT.

These components are mounted on a sliding system along a dedicated aluminum bar to prevent collisions with the magnetization system. The sensor head is height-adjustable using two support pillars with 1 cm-spaced holes, while circular clamps secure the cylindrical enclosure beneath the steel bars. A cylindrical permanent magnet near the diamond sample generates a bias field, enabling the differentiation of NV center transition lines corresponding to the four crystallographic orientations.

### 4.3. Aluminium Bar Frame

A non-magnetic, stable support structure is required to minimize mechanical vibrations and external field interference. An aluminum bar frame was selected to securely hold the steel bar, permanent magnet assembly, and NV sensor while allowing independent movement of the latter two components at a distance of approximately 10 cm beneath the steel bar. Stepper motor-driven belts control the positions of the sensor and magnet assembly. The design and configuration of the bar frame are illustrated in Figure 5, providing a detailed view of the structural layout and component arrangement. The magnet assembly is oriented with its north pole facing the −x direction.

### 4.4. Stepper Motor System

The motion system utilizes Nema 17 stepper (OMC Corporation Limited, Nanjing, China) motors controlled by TMC2209 V1.3 drivers (BIGTREETECH, Shenzhen, China) and an ESP8266 microcontroller (Amica International GmbH, Ascheberg, Germany). The motors are mounted on 3D-printed supports and drive pulleys that tension belts connected to the magnetization and sensor plates. The drivers and microcontroller are securely mounted on a 3D-printed platform attached to the aluminum frames.

### 4.5. Measurement Automation

Automation is required for magnetizing the steel bar and performing ODMR measurements. Stepper motor rotation is calibrated to travel distance using the Arduino IDE. After determining the step sizes, Python Version 3.12 scripts control the movement of both the magnetization and sensing systems. The ESP microcontroller executes position commands, ensuring precise and reproducible measurements along the steel bar.

The sensor was kept completely stationary for each measurement, which lasted 3.2 s. After this, the sensor was moved to the next position using the stepper motor. A single measurement including motor movement took 10 s, resulting in a total time of 40 min and 32 s for all 244 measurements along the steel bar.

## 5. Experimental Procedure

The objective of the experimental setup was twofold. Firstly, the collected data was essential for developing the method to vectorize the ODMR spectra. Secondly, the measurements were conducted to investigate and validate the magnetic responses associated with various influencing factors. A separate experimental setup was established to calibrate the quantum sensors to the magnetic field directions (x, y, and z).

### 5.1. Fractures and Open Bar Ends in PS Steel

To systematically investigate the magnetic response of fractures and open bar ends in prestressing steel, 140 measurements were conducted to assess the influence of varying experimental conditions on the observed ODMR spectra. The setup utilized quenched and tempered St 145/160 rods, commonly used in infrastructure from 1965 to 1978. To simulate a rupture, two bars were positioned 2 mm apart, while a single bar configuration was used for the response of open bar ends. The main focus was investigations of fracture signals, because it is the usual damage pattern in PS concrete structures. Open bar ends of PS steel only occur in rare situations, which are still important to understand. In contrast to PS steel, open bar ends of rebars exist quite frequently in PS concrete structures. The magnetic signature of both open bar ends of PS steel and rebar is quite similar; distinguishing both is an important and difficult task in building inspections.

A reference measurement was first conducted without the steel bar to capture the bias magnetic field. Before placement in the setup, the steel bars were demagnetized. The steel was then magnetized by uniformly moving the permanent magnet assembly along the +x direction and subsequently removed for measurements. ODMR measurements were taken at 1 cm intervals to evaluate the magnetic response of loose ends and complete fractures. The process was then repeated with the magnet moved in the −x direction, and its response was consecutively measured.

The experimental study involved three key aspects:Distance-dependent responses—Evaluating magnetic responses at various magnet-to-steel and sensor-to-steel bar distances.Direction of magnetization—Magnetizing the steel bar by moving the magnet in both +x and −x directions.Misalignment effects—Displacing the steel bars in the y-direction (aluminum frame width) to simulate practical misalignment scenarios.

In this setup, the vertical distance is defined as the z-direction spacing between the top of the NV sensor enclosure and the bottom of the steel bar. The sensor-to-magnet distance corresponds to the spacing between the surface of the permanent magnet assembly and the bottom of the steel bar. The NV centers are centrally positioned within the enclosure, 8.75 mm from its outer edge. Magnetic field responses were measured for steel bar displacements ranging from −3 cm to 3 cm relative to the frame center. Y-direction displacements simulate real-world scenarios where perfect sensor alignment is not achievable.

Due to the extensive number of measurements, ODMR spectra were primarily analyzed near rupture positions, with a limited number of spectra recorded per setting. An overview of the experimental parameters and their variations is provided in Table 1. For the analysis of fracture responses these parameters were varied in 121 measurements. For the open bar ends only the direction of magnetization and the displacement in y-direction were varied in 19 measurements.

### 5.2. Calibration

Vector magnetic field computation is crucial for interpreting the magnetic response of PS steel. Although shifts in ODMR transition lines provide characteristic field profiles, calculating the full vector field enables a more comprehensive physical analysis. A Cartesian coordinate system is defined as follows:The x-axis is aligned axially to the steel bar.The z-axis points upward.The y-axis follows the right-hand rule.

This coordinate system is illustrated in Figure 5, showing its relation to the aluminum bar frame.

To ensure accurate calibration, the process is conducted without the steel bar, preventing interference. An additional permanent magnet is used to maintain visibility of all 24 ODMR transition lines. A Helmholtz coil is employed to generate controlled magnetic fields along the x-, y-, and z-axes. The field strengths are precisely adjusted using a power supply, and the corresponding field values are measured using an MMC56 × 3 anisotropic magnetoresistive sensor.

ODMR spectra are recorded at varying field strengths, and transition frequencies are extracted by identifying the negative peaks (dips) in the spectra. Frequency shifts are plotted against the applied magnetic fields and linearly fitted to calculate sensitivities in MHz/G for each of the 24 transition lines. Although NV centers have four crystallographic orientations, only three are required to compute the vector field, so specific transition lines are selected for analysis.

The vector magnetic field components *B_x_*, *B_y_*, and *B_z_* are calculated by solving the following system of three equationsΔ*m_i_* = *s_x,i_B_x_* + *s_y,i_B_y_* + *s_z,i_B_z_*(1)Δ*m_j_* = *s_x,j_B_x_* + *s_y,j_B_y_* + *s_z,j_B_z_*(2)Δ*m_k_* = *s_x,k_B_x_* + *s_y,k_B_y_* + *s_z,k_B_z_*(3)
where Δ*m* denotes the transition line frequency shift, and *s_x_*, *s_y_*, *s_z_* are the field-dependent slopes of the linear fit.

## 6. Experimental Results

### 6.1. Raw Data

The raw data of a measurement is constituted by a CW-ODMR spectrum, as illustrated in Figure 6a. The spectrum displays four pairs of broader dips symmetrically positioned around the central frequency of 2.87 GHz, corresponding to the four crystallographic orientations of the NV centers. Each broader dip contains three narrower dips, representing additional energy levels resulting from the interaction with the nuclear spin-1 of the nitrogen atom, which is called hyperfine interaction. Consequently, the spectrum exhibits a total of 24 transition lines.

ODMR measurements are conducted at each position along the steel bar with a spatial resolution of 1 cm. Each measurement inherently provides two axes of information through the frequencies of the microwave and their corresponding fluorescence intensities. To visualize the spatial evolution of the magnetic response along the x-direction, a heat map plot is utilized. This plot represents the normalized ODMR response across the entire sensor range, ensuring comparability between different measurement positions. A representative heat map is included in Figure 6b. The dashed horizontal red line marks the rupture position at 127 cm, serving as a reference for interpreting localized magnetic field variations. It can be seen that the corresponding MW frequencies of the transition lines change when there is a change in the magnetic field due to the fracture in the PS steel.

### 6.2. Vectorization

Several steps of data processing are necessary to extract the magnetic field in x-, y-, and z-direction:Finding all dips in the ODMR spectra to extract the MW transition frequencies of the linesComputing the frequency shifts of the lines compared to the zero-field reference measurementForming sets of three lines each corresponding to distinct NV axis orientationsSolving the system of Equations (1)–(3) to obtain the magnetic field vectors

Python scripts were written to automate the process of vectorization. First, all the dips have to be found in each ODMR spectrum. Subsequently, the frequency shifts of the lines compared to the zero-field reference measurement are computed and slightly smoothed with a spline. Figure 7a shows the result of this process. In this example the transition lines are separated across the whole measurement length. In cases when the magnetic field is stronger they can also cross and overlap. Extensive exceptions had to be included in the algorithms to reliably find the transition lines. After this, the MW frequency of the reference measurement is subtracted and sets of three transition lines are formed. Figure 7b shows one possible set after subtracting the reference measurement. It shows the change of the transition lines in the crystallographic orientations of the diamond. For computing the vector magnetic field 5 different sets were calculated and compared for each measurement.

To convert the crystallographic orientations to the directions x, y and z, which were defined in Figure 5. The system of Equations (1)–(3) has to be solved. The result of this process is illustrated in Figure 8 for typical fracture signals.

### 6.3. Fractures in PS Steel

In total 121 measurements of fractures in PS steel were performed to analyze the behavior of the ODMR spectra. Figure 8 shows two examples of vectorized fracture signals, where a, b and c belong to a measurement after magnetization in +x direction. While d, e and f belong to a measurement after magnetization in −x direction. The directions of the poles of the magnet were unchanged with the north pole facing −x. The fracture amplitudes *B_f_* are defined for each direction. Both measurements were conducted after magnetizing the PS steel bar at a distance of 7.7 cm. The sensor distance was 11.7 cm and the displacement in y-direction −3 cm.

The data was used to develop and refine the method of vectorization. Many difficult edge cases were discovered and solutions could be implemented in the algorithms. Apart from that it was possible to extract information about the influencing of the distance of the magnet and sensor, the direction of the magnetization and the displacement in y-direction. Figure 9 shows an overview of the fracture amplitudes *B_f_* under varying boundary conditions.

Generally, the *B_z_* component has the largest fracture amplitude, followed by *B_x_* and *B_y_*. Increasing the distance of the sensor generally decreases the fracture amplitudes. At 15.7 cm the amplitudes are small with *B_x_* = 0.15 G, *B_y_* = 0.05 G and *B_z_* = 0.45 G, but the fracture is still clearly visible in the data. Increasing the distance of the magnet from 7.7 cm to 10.7 cm reduces the amplitude of the fracture by an average of 45% for *B_x_*, 24% for *B_y_* and 28% for *B_z_*. Displacing the bar ±3 cm in y-direction does not influence the fracture amplitudes of *B_x_* and *B_z_*. If the bar is completely aligned with the sensor, *B_y_* approaches zero. Increasing the displacement in the investigated range of ±3 cm the fracture amplitudes of *B_y_* increase almost linearly. If the PS steel bar is displaced in the direction +y it is characterized by a minimum followed by a maximum. If it is displaced in direction -y it is a maximum followed by a minimum. This implies that it could be possible to improve the localization of fracture signals by analyzing this component. The direction of the magnetization does not have an influence on the *Bo1* represents the magnetic. But it influences the shape of the fracture signals. They are slightly moved in the direction of travel, which is illustrated in Figure 8, where (a), (b), and (c) are magnetized moving the magnet in +x direction and (d), (e) and (f) are magnetized moving the magnet in −x direction.

### 6.4. Open Bar Ends

In addition to fracture analysis, open bar ends were investigated. A total of 19 measurements were conducted using a single PS steel bar, where both ends were visible in the measurement data. Figure 10 presents two example measurements, both taken at a measurement distance of 7.7 cm, a sensor distance of 11.7 cm, and a displacement of −3 cm. In Figure 10a–c correspond to magnetization in the +x direction, while Figure 10d–f correspond to magnetization in the −x direction.

The amplitudes of the open bar ends are defined in Figure 10, here *B_o_*_1_ represents the magnetic response at the bar’s starting point and *B*_*o*2_ at the endpoint. These responses are analyzed separately to evaluate the effect of consecutive magnetizations, first in the +x direction and subsequently in the −x direction.

The response of *B*_*o*2_ is noticeably stronger after magnetization in the +x direction, whereas *B_o_*_1_ shows a relatively weak response. In all directions, the signal is clearly visible, with *B_z_* exhibiting the strongest response, followed by *B_x_*, and *B_y_* showing the weakest. After an additional magnetization in the −x direction, both *B_o_*_1_ and *B_o_*_2_ become fully developed and appear similar, though *B_o_*_2_ is slightly weakened by the additional magnetization.

Figure 11 provides an overview of the statistical analysis, considering displacements of ±3 cm and magnetization in the ±x direction. Displacing the bar by ±3 cm in the y-direction does not affect the *B_z_* amplitudes at the open bar ends, while *B_x_* shows a slight decrease at ±3 cm. Increasing the displacement from ±2 cm to ±3 cm reduces the amplitudes by an average of 4.9%. The behavior of *B_y_* follows a similar trend to the fracture response. The additional magnetization in −x direction reduces the amplitudes of *B_o_*_2_ by an average of 30.6%, resulting in *B_o_*_2_ being 11.8% smaller than *B_o_*_1_ after the second magnetization. The influence of the magnetizing direction suggests the potential for distinguishing fractures and open bar ends through the analysis of data derived from distinct stages within the magnetization process.

## 7. Conclusions

This study aimed to evaluate the feasibility of quantum sensors in MFL testing and develop a method for processing CW-ODMR data into vectorized magnetic field data in the x, y, and z directions. The method involves:Calibrating the quantum sensor using Helmholtz coils and a separate magnetic field sensor, creating calibration curves for each direction.Performing a reference measurement in the absence of external magnetic fields.Detecting and observing the evolution of transition lines in the ODMR spectrum, measured along the length of the steel bar.Computing the frequency shifts of the lines compared to the zero-field reference measurement.Solving a system of equations to transform crystallographic sensor orientations into the calibrated directions and converting the original microwave frequency and fluorescence data into vectorized magnetic flux density.

In accordance with the specified procedure, 121 fracture measurements and 19 open-bar-end measurements were conducted under varying boundary conditions. These measurements validated the method and allowed a statistical analysis of signals in MFL testing.

The results demonstrate that quantum sensors effectively detect fractures, with *B_z_* exhibiting the strongest response, followed by *B_x_* and *B_y_*. Increasing the sensor distance reduces fracture amplitudes, but fractures remain clearly visible even at 15.7 cm, suggesting potential for detection at even greater distances. Increasing the permanent magnet assembly distance from 7.7 cm to 10.7 cm reduced fracture amplitudes by an average of 33.6% across all field directions. The direction of magnetization does not affect fracture amplitudes but slightly shifts signal positions, an effect that can be systematically accounted for in future MFL methodologies. Misalignments of the sensor in the y-direction (±3 cm) do not affect the fracture amplitudes *B_f,x_* and *B_f,z_*, while fracture amplitudes *B_f,y_* increase with displacement and invert based on displacement direction. This effect suggests the potential for improved localization of fractures.

In the case of open bar ends, the displacement behavior exhibited a pattern analogous to that of fractures; however, the effect of the direction of magnetization proved more nuanced. The open bar end that had last experienced excitation of the magnet demonstrated the strongest response. Conversely, following subsequent magnetization in the opposite direction, the response of the first magnetization remained, albeit diminished by an average of 30.6%. Analyzing the influence of magnetization direction can thus potentially distinguish fractures from open bar ends.

This study makes several important contributions to the field of NDT. It establishes a novel data processing method for quantum sensors based on NV centers, enabling CW-ODMR data to be vectorized and interpreted for fracture detection in PS steel. Additionally, it provides a statistical analysis of quantum sensor behavior under varying boundary conditions, demonstrating their effectiveness in detecting fractures with high sensitivity. The findings highlight the potential for quantum sensors to complement or replace traditional magnetic field sensors in MFL testing, offering higher sensitivity at room temperature and vector field measurement. Furthermore, this study identifies a path forward for integrating quantum sensors into automated inspection systems, paving the way for more efficient and accurate large-scale MFL testing.

Although this study applied a wide range of boundary conditions, the experimental setup was relatively simple compared to real-world PS concrete structures. Notably, the influence of rebars was not investigated, which is a critical factor in practical MFL testing. Rebars introduce additional magnetic fields, which impact fracture detection and signal interpretation. Furthermore, while the vectorization process successfully transformed CW-ODMR measurements into magnetic field data, further validation is required in real-world environments to confirm its robustness.

To further develop the application of quantum sensors in MFL testing, future research should:Investigate the influence of rebars in PS concrete structures and develop methods to suppress interfering magnetic signals.Refine data processing techniques by integrating machine learning or advanced signal processing algorithms to improve fracture classification and localization.Develop hybrid sensor setups, combining quantum and conventional sensors to allow for a direct and objective comparison in practical NDT applications.Validate the method in more complex laboratory setups and eventually in real-world building inspections, ensuring robustness under environmental conditions.

This study demonstrates that quantum sensors based on NV centers can be successfully applied to MFL testing for fracture detection in PS steel. Their unique properties offer a promising alternative to conventional sensors. However, further research is required to optimize data processing, environmental robustness, and large-scale applicability. If these challenges are addressed, quantum sensors could revolutionize MFL-based NDT applications, significantly improving fracture detection and reducing false positives. Beyond MFL testing, their unique capabilities could enable a wide range of applications. While their immediate impact lies in building inspections, their potential extends far beyond NDT, opening new possibilities for advanced sensing technologies in various scientific and industrial fields.

## Figures and Tables

**Figure 1 sensors-25-07279-f001:**
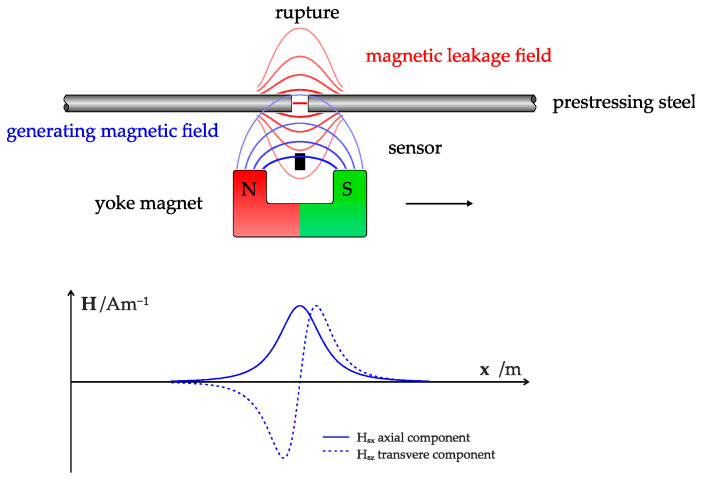
Sketch of the process for measuring the magnetic field and resulting stray field of the fracture in the PS steel.

**Figure 2 sensors-25-07279-f002:**
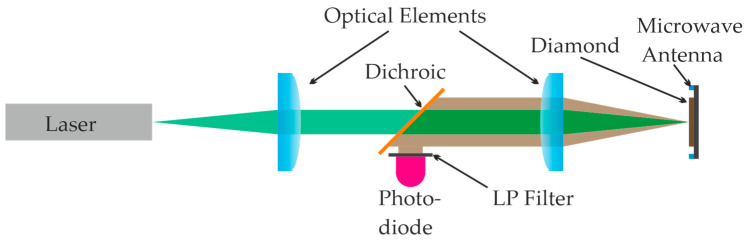
Scheme of optical system embedded in the sensor head.

**Figure 3 sensors-25-07279-f003:**
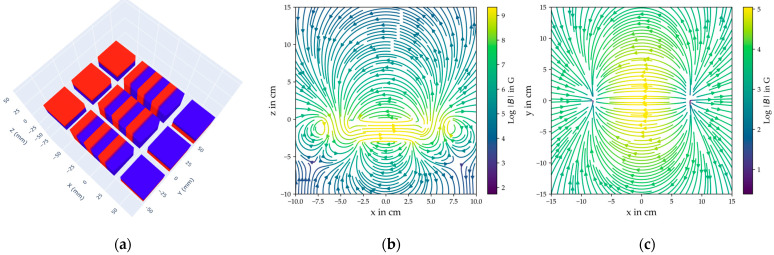
Simulation results of the B-field spatial profile generated by the permanent magnet assembly. (**a**) illustrates the geometric configuration of the neodymium permanent magnets, with red representing the south pole and blue representing the north pole of the individual magnets. In both (**b**,**c**), the colour bar is employed to display the logarithm of the field’s magnitude. (**b**) The magnetic field profile in the y = 0 cm plane is demonstrated here. This corresponds to a lateral perspective from the centre of the magnet arrangement. (**c**) Magnetic field profile in the z = 10 cm plane. This corresponds to a top view of the magnet from a distance of 10 cm.

**Figure 4 sensors-25-07279-f004:**
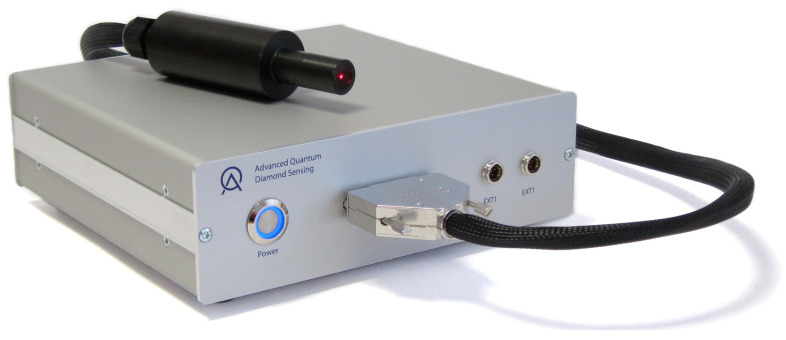
The NV sensor, manufactured at Advanced Quantum GmbH. The black cylindrical enclosure (sensor head) contains the NV centers, the green laser, the MW source, the photodetectors, the data acquisition system and additional optics. The diamond is positioned at the rear end of the sensor head. A wire connects the sensor head to the sensor box. Part of the red photoluminescence from the NV centers is visible through the transparent wall at the end of the cylindrical enclosure.

**Figure 5 sensors-25-07279-f005:**
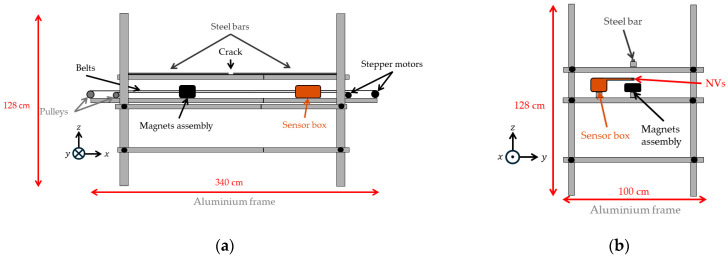
Schematics and photos of the experimental setup design. The distance of the sensor box and magnetic assembly to the steel bar is adjustable. (**a**) Schematic length side view. (**b**) Schematic width side view. (**c**) Photo length side view. (**d**) Photo width side view.

**Figure 6 sensors-25-07279-f006:**
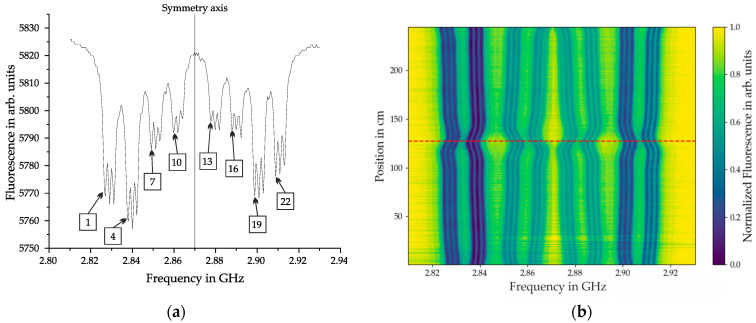
(**a**) CW-ODMR spectrum showing transition lines for all four crystallographic orientations of NV centers. Four pairs of fluorescence dips are symmetrically positioned around the 2.87 GHz frequency. Each pair contains three narrower dips due to additional energy levels from the nuclear spin-1 interaction of the nitrogen atom (mS = 0, ±1). The dips are numbered from 1 to 24 in ascending order of transition frequency. (**b**) Normalized ODMR response along the x-direction, illustrating fluorescence variations across the sensor’s position range. Measurements were taken at a vertical distance of 11.7 cm below the steel bars following magnetization in +x direction with a distance of 7.7 cm. The dashed red line at 127 cm indicates the rupture position.

**Figure 7 sensors-25-07279-f007:**
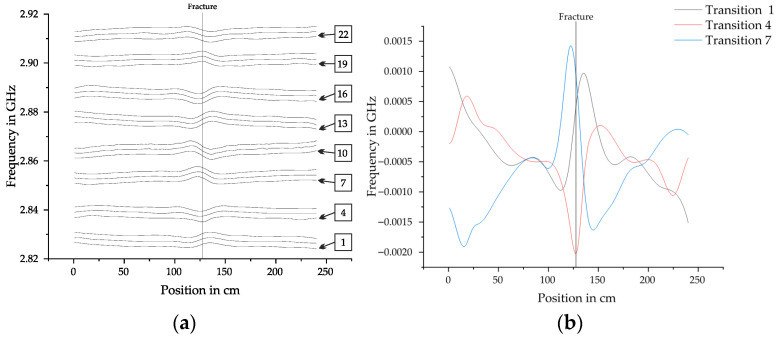
(**a**) All 24 transition lines with their corresponding frequencies at each position along the measurement length with a complete fracture of the prestressing steel at 127 cm. (**b**) Set of transition lines 1, 4 and 7 after subtracting the frequencies of the reference measurement. They show the magnetic field in the crystallographic orientations of the diamond.

**Figure 8 sensors-25-07279-f008:**
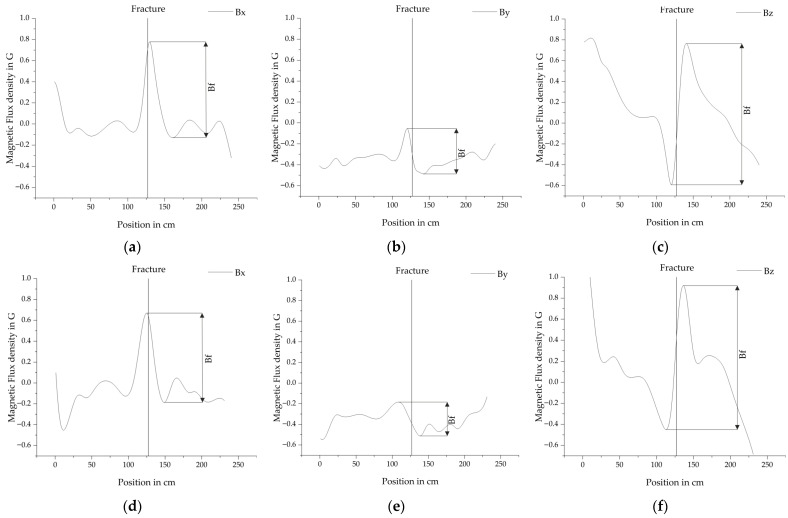
Vectorized fields of the measurement with the frequencies converted to magnetic flux density in Gauss. For further statistical analysis the amplitudes of the fracture are defined as *B_f_* for (**a**) the magnetic field in x-direction after magnetization in +x direction, (**b**) the magnetic field in y-direction after magnetization in +x direction, (**c**) the magnetic field in z-direction after magnetization in +x direction, (**d**) the magnetic field in x-direction after magnetization in −x direction, (**e**) the magnetic field in y-direction after magnetization in −x direction and (**f**) the magnetic field in z-direction after magnetization in −x direction. Both measurements were conducted after magnetizing the PS steel bar at a distance of 7.7 cm. The sensor distance was 11.7 cm and the displacement in y-direction −3 cm.

**Figure 9 sensors-25-07279-f009:**
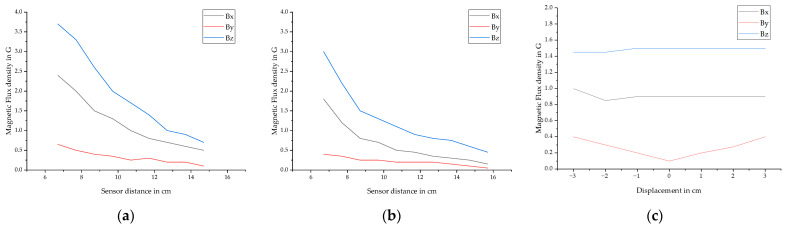
Overview of the influence of varying boundary conditions on fracture amplitudes in x, y and z direction where (**a**) shows the influence of the sensor distance after magnetizing the PS steel at a distance of 7.7 cm. (**b**) shows the influence of the sensor distance after magnetizing the PS steel at a distance of 10.7 cm and (**c**) shows the influence of displacement in y-direction. The sensor distance was kept at a constant of 11.7 cm after magnetizing the PS steel at a distance of 7.7 cm.

**Figure 10 sensors-25-07279-f010:**
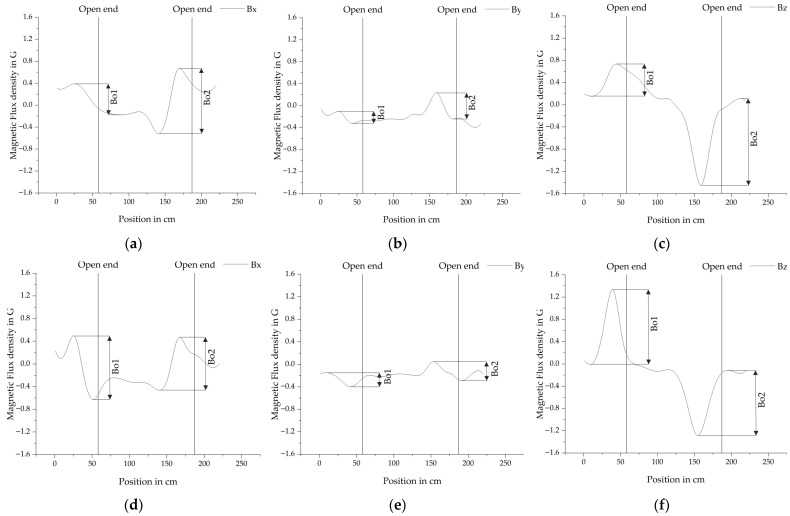
Vectorized fields of the measurement with the frequencies converted to magnetic flux density in Gauss. For further statistical analysis the amplitudes of the open bar ends are defined as *B_o_* for (**a**) the magnetic field in x direction after magnetization in +x direction, (**b**) the magnetic field in y-direction after magnetization in +x direction, (**c**) the magnetic field in z direction after magnetization in +x direction, (**d**) the magnetic field in x-direction after magnetization in −x direction, (**e**) the magnetic field in y direction after magnetization in −x direction and (**f**) the magnetic field in z direction after magnetization in −x direction. The sensor distance was a constant of 11.7 cm after magnetizing the PS steel at a distance of 7.7 cm.

**Figure 11 sensors-25-07279-f011:**
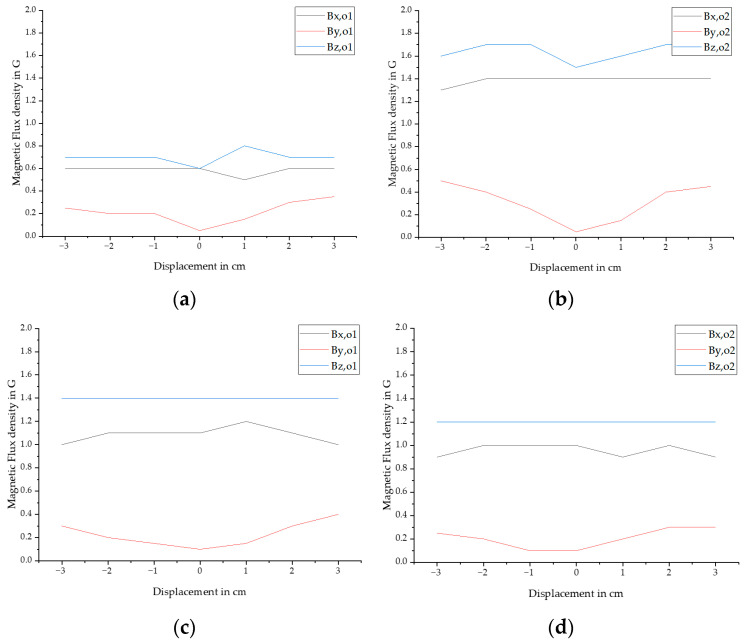
Overview of the influence of the displacement in y direction on magnetic response amplitudes in x, y and z direction where (**a**) shows the influence on *B_o_*_1_ after magnetizing in +x direction, (**b**) shows the influence on *B_o_*_2_ after magnetizing in +x direction (**c**) shows the influence on *B_o_*_1_ after magnetizing in −x direction, (**d**) shows the on *B_o_*_2_ after magnetizing in −x direction. The sensor distance was a constant of 11.7 cm after magnetizing the PS steel at a distance of 7.7 cm.

**Table 1 sensors-25-07279-t001:** Overview of experimental parameters and their respective variations used in the study.

Parameter	Variations
Magnet distance to PS steel	7.7 cm and 10.7 cm
Sensor distance to PS steel	6.7 to 15.7 cm in steps of 1 cm
Direction of magnetization	−x and +x
Displacement in y-direction	−3 to 3 cm in steps of 1 cm

## Data Availability

The original contributions presented in this study are included in the article. Further inquiries can be directed to the corresponding author.

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
