# Peer review of "Implementing Nitrogen Vacancy Center Quantum Sensor Technology for Magnetic Flux Leakage Testing"

_sensors, 2025, doi:10.3390/s25237279_

Round 1

Reviewer 1 Report

Comments and Suggestions for Authors

The manuscript "Implementing Quantum Sensor Technology for Magnetic Flux Leakage Testing" addresses a specific practical problem and should be evaluated based on its practical utility.

Regarding the second aspect of any scientific publication, namely, scientific novelty, I believe it is rather weak in this case. Essentially, the authors used the well-known MFL method, replacing the sensors typically used in this method with a commercially available sensor from NV centers (the manufacturer, Advanced Quantum GmbH, is listed as an affiliation). The authors then conducted extensive and undoubtedly useful work integrating this sensor into the MFL method.

Overall, I believe the article deserves publication precisely because of its practical significance, not its scientific novelty.

I nevertheless have a number of comments and questions for the authors.

1. While listing known sensor types (L67-88), the authors completely overlook the most suitable sensor type for their task: an alkali metal vapor optical sensor. The Introduction to the manuscript gives the impression that the NV sensor is the only type of sensor using ODMR, although in fact, it is only one in a vast class of such sensors. Compact sensors using ODMR in Cs or Rb exhibit approximately 100 times better sensitivity than NV sensors, with a sensing element size of a few millimeters. They mostly measure the field modulus, but a number of schemes exist that allow vector measurements with them. Furthermore, the advantage of vector measurements over scalar ones in the MFL method remains to be proven.

As is well known, NV CW-ODMR vector sensors are characterized by rather low sensitivity—on the order of nT in a 1 Hz bandwidth. Furthermore, they require a large bias magnetic field to operate. These shortcomings are offset by their sole advantage—high spatial resolution (<0.1 mm). This advantage is not utilized in the MFL method at all, as typical distances to objects are 10 cm or more, and all records (Fig. 7) are characterized by precisely this spatial resolution. Given this, it is necessary to justify the use of the NV sensor in the MFL method by supplementing the introduction with information on Cs- and Rb-based sensors.

2. Obtaining a single spectrogram, similar to the one shown in Fig. 5a, requires a relatively long time, determined by the longitudinal relaxation time T1. This time is typically tens of milliseconds, so recording a full spectrum with 24 peaks takes over a second (sometimes orders of magnitude longer). Accordingly, compiling a two-dimensional spectrogram similar to Fig. 5b should take hundreds of seconds. Unless I'm mistaken, the authors completely ignore this aspect in the manuscript.

3. The description of the operating principles of the NV sensor given in L213-232 cannot be considered satisfactory. NV centers do not necessarily need to be excited by light at 432 nm, and they also emit light at wavelengths other than 637 nm. Calling the second decay pathway "phosphorescence" is most likely incorrect, since it is nonradiative decay.

4. L393 "additional energy levels resulting from the interaction with the nuclear spin of the nitrogen atom" – this interaction is called the hyperfine interaction.

5. L78 "include flux gates"… "A fluxgate magnetometer" is usually written together.

6. L74 "Δm_i =" variables denoted by Latin letters, with the exception of vectors, should be written in italics.

7. L206: "All the necessary electronics are included in the control unit with the following elements" – the list is missing an analog transimpedance amplifier (photocurrent amplifier).

Author Response

Thank you for your detailed descriptions and your valuable input. I made several changes in the manuscript according to your review.

Question / Task 1: 

  1. While listing known sensor types (L67-88), the authors completely overlook the most suitable sensor type for their task: an alkali metal vapor optical sensor. The Introduction to the manuscript gives the impression that the NV sensor is the only type of sensor using ODMR, although in fact, it is only one in a vast class of such sensors. Compact sensors using ODMR in Cs or Rb exhibit approximately 100 times better sensitivity than NV sensors, with a sensing element size of a few millimeters. They mostly measure the field modulus, but a number of schemes exist that allow vector measurements with them. Furthermore, the advantage of vector measurements over scalar ones in the MFL method remains to be proven.
    As is well known, NV CW-ODMR vector sensors are characterized by rather low sensitivity—on the order of nT in a 1 Hz bandwidth. Furthermore, they require a large bias magnetic field to operate. These shortcomings are offset by their sole advantage—high spatial resolution (<0.1 mm). This advantage is not utilized in the MFL method at all, as typical distances to objects are 10 cm or more, and all records (Fig. 7) are characterized by precisely this spatial resolution. Given this, it is necessary to justify the use of the NV sensor in the MFL method by supplementing the introduction with information on Cs- and Rb-based sensors.

Answer 1:

Thank you for pointing out that I overlooked the optical sensor for alkali metal vapours in my description of alternative sensor types. This sensor type should be examined more closely for NDT applications. The sensitivity is impressive and it is really interesting that it can also provide vector field measurements. For fracture detection in prestressing steel they might exhibit limitations in terms of dynamic range. In our laboratory setup the highest field was around 3.5 G. They may have limitations in terms of their dynamic range for fracture detection in prestressed steel. In our laboratory setup, the highest field was around 3.5 G. When magnetising steel in real-world applications, there are sometimes reinforcement arrangements that cause magnetic fields of up to 30 G. But for other magnetisation schemes, optical sensors using alkali metal vapour could be a great option and deserve more research in the field of non-destructive testing and building inspections.

Revised text (line 84 to 101):
Superconducting Quantum Interference Device (SQUID) magnetometers offer superior sensitivity in the aT/rtHz range with minimal hysteresis but necessitate cryogenic cooling to approximately 4 K (low-Tc) or 77 K (high-Tc) depending on design, limiting their field applications. Requiring a cryogenic engine also significantly increase the price [21]. Opti-cally pumped magnetometers (OPMs) in alkali-metal vapors such as cesium or rubidium. These alkali metal vapor optical sensors achieve sensitivities in the range of fT/√Hz at room temperature with millimeter-scale sensing volumes [25]. While most OPMs are sca-lar devices that measure the magnetic-field modulus, vector-measurement schemes have been demonstrated [26]. An additional option is the use of multi-channel systems [25]. Their advantages include high sensitivity and compactness; however, their dynamic range and gradient tolerance is limited [27, 28, 29]. The price of a fully functioning sensor setup is higher than conventional sensors, but still more affordable than SQUID magne-tometers [25]. Quantum sensors based on negatively charged nitrogen vacancy (NV) cen-ters achieve sensitivities from pT/√Hz to nT/√Hz and also have high dynamic range up to several mT. In addition, they enable simultaneous measurement of magnetic fields in multiple directions, offering a promising alternative for advanced MFL testing. They oper-ate at room temperature and exhibit low hysteresis. They are also compact and robust, enabling their use on construction sites [24, 30].

Question / Task 2
2. Obtaining a single spectrogram, similar to the one shown in Fig. 5a, requires a relatively long time, determined by the longitudinal relaxation time T1. This time is typically tens of milliseconds, so recording a full spectrum with 24 peaks takes over a second (sometimes orders of magnitude longer). Accordingly, compiling a two-dimensional spectrogram similar to Fig. 5b should take hundreds of seconds. Unless I'm mistaken, the authors completely ignore this aspect in the manuscript.

Answer:
The sensor was kept completely stationary for each measurement, which lasted 3.2 seconds. After this, the sensor was moved to the next position using the stepper motor. A single measurement including motor movement took 10 seconds, resulting in a total time of 40 minutes and 32 seconds for all 244 measurements along the steel bar. This is far too long for practical applications, but we hope to develop a technical solution to perform measurements while moving the sensor and obtaining the position from a rotary encoder.
I also included this information in the article under 4.5 Measurement automation.

Revised text (line 345 to 348)

The sensor was kept completely stationary for each measurement, which lasted 3.2 seconds. After this, the sensor was moved to the next position using the stepper motor. A single measurement including motor movement took 10 seconds, resulting in a total time of 40 minutes and 32 seconds for all 244 measurements along the steel bar. 

Question / Task 3

  1. The description of the operating principles of the NV sensor given in L213-232 cannot be considered satisfactory. NV centers do not necessarily need to be excited by light at 432 nm, and they also emit light at wavelengths other than 637 nm. Calling the second decay pathway "phosphorescence" is most likely incorrect, since it is nonradiative decay.

Answer

I revised the passage to incorporate your valid critique.

Revised Text

"We use a 525 nm green laser to optically pump the NV centers into excited states. The system can then decay back to the ground state through two pathways. The first one is the fluorescence pathway, where the electrons rapidly relax to the ground state within nanoseconds. The emission shows a zero-phonon line (ZPL) at 637 nm. Additionally, the second pathway involves intersystem crossing (ISC) to a metastable singlet state, leading to an effectively non-radiative decay process. The resulting photons are detected by a photodiode, with optical filters used to separate the excitation light from the fluorescence photons.

Additionally, the absorption and emission spectra are influenced by lattice vibrations, known as phonons. The specific vibrational modes, or phonon modes, broaden the optical absorption and thus allow for non-resonsant excitation. In the emission, the spectrum is broadened by the phonon-side-band to a spectral range between 630nm to 750nm.

When the laser is activated, fluorescence produces a steady stream of photons. By continous excitation, the state is initialized by optical cycling into the bright state. Optical filters ensure that only fluorescence photons within the 600 to 800 nm range are detected by the photodiode. If the quantum state is manipulated to favor ISC, fewer photons are detected. This manipulation is achieved using a MW signal.

A continuous sweep of MW frequencies is applied at the NV center's location, typically centered at 2.87 GHz, generating a symmetric spectrum around this frequency. The MW frequency range determines the maximum observable magnetic field magnitudes. Light emission is recorded for each MW frequency, forming the Optically Detected Magnetic Resonance (ODMR) spectrum. This technique, known as continuous-wave ODMR (CW-ODMR), involves the continuous application of both the optical excitation laser and MW signals to the NV centers”

Question / Task 4

  1. L393 "additional energy levels resulting from the interaction with the nuclear spin of the nitrogen atom" – this interaction is called the hyperfine interaction.

Revised text

“Each broader dip contains three narrower dips, representing additional energy levels re-sulting from the interaction with the nuclear spin-1 of the nitrogen atom, which is called hyperfine interaction.”

Question / Task 5

  1. L78 "include flux gates"… "A fluxgate magnetometer" is usually written together.

Revised text

“More sensitive alternatives include fluxgate magnetometers, magnetoresistive (MR), and Giant-magnetoresistive (GMR) sensors, which provide sensitivity in the pT/rtHz range [17].”

Question / Task 6

  1. L74 "Δm_i =" variables denoted by Latin letters, with the exception of vectors, should be written in italics.

Answer

I have corrected this in the complete manuscript.

Question / Task 7

  1. L206: "All the necessary electronics are included in the control unit with the following elements" – the list is missing an analog transimpedance amplifier (photocurrent amplifier).

Answer

You are correct. The sensor head contains a transimpedance amplifier. The amplified signal, converted from photocurrent to voltage, is then sent to the control box. I updated the list in the article.

Reviewer 2 Report

Comments and Suggestions for Authors

A very thorough experimental study into the application of quantum sensors for MFL / remanent magnetisation measurements. Some minor comments below.

Sections and subsections should be numbered.

Magnetic fields are measured in Gauss. SI units would be preferred, but not essential.

Page 2: “the MFL signal can be measured either during magnetization or in remanence [14, 15].” This is slightly misleading. It seems that industrial systems for NDT of PS concrete almost exclusively use the remanent magnetisation technique. Some clarification would be welcome.  

Page 2: “State-of-the-art laboratory Hall-proves” should maybe read “State-of-the-art laboratory Hall-probes”?

Page 5: “The required magnetizing field at the steel bar's position is approximately 100 G at a vertical distance of 10 cm, corresponding to typical conditions where steel bars are embedded 10 cm beneath a concrete surface.” It is not clear what this means. Does the 100 G refer to the field at the poles of the magnet or at the steel bar’s position.

Page 6: The ‘geometric arrangement of the magnetic components’ is difficult to visualise. A diagram would be welcome.

Figure 3: On what plane is this spatial distribution? At the poles of the magnet or at the target object?

How long did each measurement take?

An experimental comparison with a traditional Hall sensor, maybe just a single scan in the same rig, would have helped to highlight the advantages of this technique. Have the authors undertaken any comparison tests and, if they have, would it be possible to include some results here?

Author Response

First of all, I would like to thank you for taking the time to improve my article. I agree with almost all of your comments and have tried to revise my manuscript accordingly.

Question / Task 1

Sections and subsections should be numbered.

Answer

Thank you for that comment. I have added numbering to all headings.

Question / Task 2

Magnetic fields are measured in Gauss. SI units would be preferred, but not essential.

Answer

Thank you for the remark — I appreciate your perspective on using SI units. However, I would prefer to keep the values in Gauss, as this unit matches the typical range of our measurements more naturally and provides better readability for the specific field strengths we are dealing with.

Question / Task 3

Page 2: “the MFL signal can be measured either during magnetization or in remanence [14, 15].” This is slightly misleading. It seems that industrial systems for NDT of PS concrete almost exclusively use the remanent magnetisation technique. Some clarification would be welcome.  

Answer

Thank you for this insight. I have revised the text to make it clearer that all available options provide meaningful data and are used with great systems.

Revised text

The magnet can be either a permanent magnet or an electromagnet, and the MFL signal can be measured either during magnetization or in remanence [14, 15]. Both measurement options are used in industrial applications [16]. Some systems only measure during magnetization, which is mainly the case with permanent magnet configurations [17, 18]. Other systems rely primarily on remanent measurements [9] and others combine both approaches [13, 15].

Question / Task 4

Page 2: “State-of-the-art laboratory Hall-proves” should maybe read “State-of-the-art laboratory Hall-probes”?

Revised Text (line 72 to 73)

State-of-the-art laboratory Hall-probes can reach up to nT/√Hz range at room temperature.

Question / Task 5

Page 5: “The required magnetizing field at the steel bar's position is approximately 100 G at a vertical distance of 10 cm, corresponding to typical conditions where steel bars are embedded 10 cm beneath a concrete surface.” It is not clear what this means. Does the 100 G refer to the field at the poles of the magnet or at the steel bar’s position.

Revised text (line 268 to 269)

The target magnetizing field at the PS steel bar (10 cm below the concrete surface) is approximately 100 G.

Question / Task 6

Page 6: The ‘geometric arrangement of the magnetic components’ is difficult to visualise. A diagram would be welcome.

Answer

I added another figure to illustrate the geometric arrangement figure 3 (a) on page 7.

Question / Task 7

Figure 3: On what plane is this spatial distribution? At the poles of the magnet or at the target object?

Answer

Figure 3 (b) shows the plane, where y = 0. This means that it is like a cut through the middle of the magnet. The z direction shows the distance to the steel. This corresponds to the drawing in figure 4. Which means that the two poles of the magnet are at z = 0 and x = -5 / +5. The steel bar would typically be at z = 10 cm. Figure 3 (c) shows something like a top view onto the magnet.

I have also added two sentences to the description to make it easier to visualise.

Revised text

Figure 3: Simulation results of the B-field spatial profile generated by the permanent magnets assembly. (a) illustrates the geometric configuration of the neodymium permanent magnets, with blue representing the south pole and blue representing the north pole of the individual magnets. In both (b) and (c), the colour bar is employed to display the logarithm of the field's magnitude. The corresponding coordinate system is also shown in Figure 5. (b) The magnetic field profile in the y = 0 cm plane is demonstrated here. This corresponds to a lateral perspective from the centre of the magnet arrangement. (c) Magnetic field profile in the z = 10 cm plane. This corresponds to a top view of the magnet from a distance of 10 centimetres.

Question / Task 8

How long did each measurement take?

Answer

The sensor was kept completely stationary for each measurement, which lasted 3.2 seconds. After this, the sensor was moved to the next position using the stepper motor. A single measurement including motor movement took 10 seconds, resulting in a total time of 40 minutes and 32 seconds for all 244 measurements along the steel bar. This is far too long for practical applications, but we hope to develop a technical solution to perform measurements while moving the sensor and obtaining the position from a rotary encoder.

I also included this information in the article under 4.5 Measurement automation.

Question / Task 9

An experimental comparison with a traditional Hall sensor, maybe just a single scan in the same rig, would have helped to highlight the advantages of this technique. Have the authors undertaken any comparison tests and, if they have, would it be possible to include some results here?

Answer

This is a really important point. Which is why I included this passage in the conclusions section:

“To further develop the application of quantum sensors in MFL testing, future research should:

  • Develop hybrid sensor setups, combining quantum and conventional sen-sors to allow for a direct and objective comparison in practical NDT applications.”

We performed separate measurements outside of the setup for this study with our classic sensors only sensing Bx and included in an electromagnet. The results look similar, but it is hard to tell if we are comparing the sensor of the magnetization method. I think it is necessary to create an objective method for comparison to determine the actual quality of the measurements with different sensor types. We could have included a cheap Hall-sensor in the setup, but that doesn’t seem satisfactory. We are planning to perform high quality measurements with 3D sensors of different kinds in the future.

Reviewer 3 Report

Comments and Suggestions for Authors

In this manuscript, a quantum sensor is used for the magnetic flux leakage testing. The use of new sensors is interesting. However, major revisions are required to make the manuscript clearer to readers.

  1. What makes quantum sensors based NV center especially suitable for this particular applications. More explanations should be given.
  2. The manuscript is totally based on schematic drawing. A real photo of the experimental setup should be shown. Besides, a photo of the quantum sensor should be given to show its details.
  3. More details of the quantum sensor should be given, such as its physical dimension, sensitivity, linearity range, etc.
  4. In the title, the concept of “quantum sensor” is too broad. It is suggested to specify it, such as adding nitrogen vacancy center.
  5. From figure 4, it seems that the sensor box is far away from the magnets and crack. How are the leakage magnetic field being sensed in such distance?
  6. In the references, too many sources are in German, which is not appropriate for an international journal. Most readers of this journal will not understand them. Please cite sources in English, and consider worldwide contributions.
  7. From the results shown in figure 7, the variation of magnetic flux density is in the magnitude of 0.1 G. Such small field can be easily affected by environmental fields. How does the sensor suppress the noise caused by environmental field variations?
  8. From the results shown in figure 7, we can see that the baseline is not stable. How is the repeatability of the tests?

Author Response

Thank you very much for your feedback. I greatly appreciated your comments and believe that I was able to improve my manuscript based on them.

Question / Task 1

  1. What makes quantum sensors based NV center especially suitable for this particular applications. More explanations should be given.

Answer

I added a few more sentences to further describe why the sensor is suitable for this application. The main points are dynamic range, operation at room temperature, high sensitivity compared to conventional sensors, low hysteresis, compactness and robustness. However I think that other sensors are also good options and their application should be researched in more depth. That is why I wanted to list the sensors objectively instead of pointing too aggressively to quantum sensors based on NV centers.

Revised Text

line 96 to line 101:

Quantum sensors based on negatively charged nitrogen vacancy (NV) centers achieve sensitivities from pT/√Hz to nT/√Hz and also have high dynamic range up to several mT. In addition, they enable simultaneous measurement of magnetic fields in multiple direc-tions, offering a promising alternative for advanced MFL testing. They operate at room temperature and exhibit low hysteresis. They are also compact and robust, enabling their use on construction sites [24, 30].

line 136 to 145:

Quantum sensors based on NV centers exhibit high sensitivity while operating at room temperature. Additionally, they can simultaneously measure multiple directions of the magnetic field within a single measurement, enhancing their effectiveness in complex magnetic field analyses. Provided that the bias field is optimized for the application, they have a high dynamic range and gradient tolerance. They have been utilized in laboratory settings for some time; however, only recent technological advancements have enabled their application in real-world scenarios. As a novel sensor type in NDT measurements, they exhibit promising properties for a broad range of magnetic methods. Nevertheless, their high sensitivity and complex data structures present challenges in data interpretation and practical implementation.

Question / Task 2

  1. The manuscript is totally based on schematic drawing. A real photo of the experimental setup should be shown. Besides, a photo of the quantum sensor should be given to show its details.

Answer

Thank you for that comment. I also prefer seeing photos in addition to the schematic drawings. I added two photos of the setup in Figure 5 on page 8.

Question / Task 3

  1. More details of the quantum sensor should be given, such as its physical dimension, sensitivity, linearity range, etc.

Answer

Thank you for that request. I added a passage to 4.2 Sensing system

Revised text

“The NV sensor, developed by Advanced Quantum GmbH, consists of a sensor box and a cylindrical diamond sample enclosure. This arrangement is shown in Figure 4. The diameter of the sensor head is 35 mm, with a length of 12 mm, including the laser driver, detector, optics and TIA. The dimensions of the sensor element are 0.5 mm x 0.5 mm x 0.5 mm. Precise measurement of the sensitivity has not been conduct-ed, however, it is estimated to be within the range of 1 nT/rtHz. Linearity range: In the measurement mode with post-processing that was utilised, the capacity is constrained by the microwave source. In this system, it is approximately 7.5 mT.”

Question / Task 4

  1. In the title, the concept of “quantum sensor” is too broad. It is suggested to specify it, such as adding nitrogen vacancy center.

Answer

Good point. I have updated the title accordingly.

Question / Task 5

  1. From figure 4, it seems that the sensor box is far away from the magnets and crack. How are the leakage magnetic field being sensed in such distance?

Answer

In all measurements, the true sensor–steel distance was between 6.7 cm and 15.7 cm, as stated in Table 1 (sensor-to-steel distance). The sensor box is positioned below the steel bar, and the distance is adjusted in 1 cm steps using the height-adjustable pillars. At these distances, the remanent magnetic flux leakage fields originating from prestressing steel fractures are still on the order of 0.1–3.75 G, which lies well above the detection threshold of the NV ensemble magnetometer as long as the interfering signals of rebars are not too strong.

Question / Task 6

  1. In the references, too many sources are in German, which is not appropriate for an international journal. Most readers of this journal will not understand them. Please cite sources in English, and consider worldwide contributions.

Answer

I must apologise that many of the sources are in German. Literature on quantum sensors and background information is almost completely kept in English, but for the literature about magnetic flux leakage testing for fracture detection in prestressing steel this did not seem appropriate. For historic reasons (econstruction of infrastructure after the Second World War using prestressed steel, which is now known to be highly susceptible to hydrogen-induced stress corrosion cracking) a lot of research was conducted in Germany. Most authors have also written articles in English, so I have decided to cite the English articles and, in addition, the more comprehensive German references such as research reports or dissertations.

Question / Task 7

  1. From the results shown in figure 7, the variation of magnetic flux density is in the magnitude of 0.1 G. Such small field can be easily affected by environmental fields. How does the sensor suppress the noise caused by environmental field variations?

Answer:

At present, no measures are in place to suppress ambient noise. The Earth's magnetic field exhibits fluctuations, though these are not of a particularly large magnitude. In real-world applications, the limiting factor is typically the signals of the reinforcement bars. The present focus of our research is the development of methods that are designed to enhance the suppression of the aforementioned signals. In the context of prospective advancements, ambient noise must be a factor of paramount importance when contemplating field measurements in close proximity to magnetic sources, such as power cables. Furthermore, in scenarios where the measurement of cracks deeper in concrete is undertaken, it is imperative to consider the potential for the measured response to fall within the same range as the ambient fluctuations.

Question / Task 8

  1. From the results shown in figure 7, we can see that the baseline is not stable. How is the repeatability of the tests?

Answer 

Repeatability is very high with those static magnetic measurements. Measurements are only altered if the magnetic field is changed. The unstable baseline results from magnetic effects of the steel bar as the fracture signal does not exist without the magnetic fields of the unbroken part of the steel.

Round 2

Reviewer 3 Report

Comments and Suggestions for Authors

My comments have been well addressed.